# Refractive error among school-aged children in Bhutan: A secondary analysis of data from nationwide Bhutan School Sight Survey

Indra Prasad Sharma[1,2]*, Kovin Shunmugam Naidoo[2,3,4], Khathutshelo Percy Mashige[2,3], Nor Tshering Lepcha[1]

1 Department of Ophthalmology, Jigme Dorji Wangchuck National Referral Hospital, Thimphu, Bhutan, 2 Discipline of Optometry, School of Health Sciences, University of KwaZulu-Natal, Durban, South Africa, 3 African Vision Research Institute, University of KwaZulu-Natal, Durban, South Africa, 4 Department of Optometry, University of New South Wales, Sydney, Australia

* indrapsharma@gmail.com

## Abstract

This study aimed to determine the prevalence, demographic distribution, and predictors of refractive error (RE) among school children in Bhutan using secondary data from the 2019 Bhutan School Sight Survey (BSSS). It also sought to compare these population-level findings with standardized estimates from the Refractive Error Study in Children (RESC). Secondary data from 586 schools, comprising 164,365 school-enrolled children aged 5–18 years (94.3% of the national school-age population), were analyzed. As part of the BSSS, vision screening and cycloplegic refraction were conducted following the RESC protocol. Spectacles were provided at no cost to children requiring correction. Descriptive statistics were used to analyse data on demographic and school-related variables, and logistic regression to identify associated predictors. Findings were compared with previously published RESC data to contextualise observed trends. Visual impairment (VI) affected 12.14% (95% CI: 11.98-12.30) of school-aged children in Bhutan, with higher rates in females (13.16%) than males (11.07%) (p < 0.01). RE was the leading cause, accounting for 11.71% of all cases and 96.1% of VI. Myopia was most common (5.78%), followed by astigmatism (5.26%) and hypermetropia (0.67%). Multivariate analysis showed that female students (AOR = 1.18), those in private schools (AOR = 1.41), and urban schooling (AOR = 1.68) were significant RE predictors (p < 0.01). Of 19,243 students diagnosed with RE, only 11.5% had appropriate spectacles. Among those needing new spectacles (n = 17,021), 96.2% eventually received them within four months, while 3.8% were lost to follow-up. In Bhutan, RE affects over 10% of school children, primarily due to myopia and astigmatism. The higher burden among females, urban students, and private schools highlights persistent inequities. These results are consistent with prior RESC findings and demonstrate that large-scale school screening is both feasible and scalable. Strengthening school-based vision services, timely spectacle

**Data availability statement:** Yes - all data are fully available without restriction; All datasets are made available in the Supporting Information.

**Funding:** The author(s) received no specific funding for this work.

**Competing interests:** The authors have declared that no competing interests exist.

delivery, and cross-sectoral coordination are essential to close the gap and address uncorrected RE.

## Introduction

Uncorrected refractive error (URE) is the most readily treatable form of visual impairment (VI). However, it persists as the leading cause of VI worldwide and a major contributor to preventable vision loss among school children. The prevalence of URE, particularly myopia, has reached epidemic proportions, with the highest burden observed in East and Southeast Asia [1]. Despite the availability of effective corrective interventions, only 36% of individuals worldwide with distance VI attributable to refractive error (RE) have access to appropriate refractive correction [2].

URE has been shown to negatively affect children's academic achievement, social functioning, and overall developmental outcomes [3]. A global pooled analysis estimated that about 2.26 per 1,000 individuals under 20 in Southeast Asia have URE, indicating a notable regional burden [4]. Notably, the prevalence of myopia has demonstrated a marked upward trend, with projections indicating that over 740 million children and adolescents worldwide will be affected by 2050 [5]. In response to the increase in RE, the World Health Organization (WHO) launched the SPECS 2030 initiative, which targets a 40% increase in effective refractive error coverage (eREC) by 2030 [6]. Realisation of this target necessitates robust, evidence-informed strategies. However, current national-level data remain limited, highlighting the urgent need for comprehensive, population-based survey data [7].

Bhutan has integrated eye health within national universal health coverage priorities [8]. It is also one of the few countries to hold a workshop for the WHO SPECS 2030 initiative. Despite this, the country's mountainous geography and widely dispersed rural population have posed significant challenges to the delivery of eye care services. Prior to 2019, there was a notable absence of systematic data on RE among children. In response, the Ministry of Health, in partnership with the Ministry of Education and international stakeholders, implemented the Bhutan School Sight Survey (BSSS) in 2019. The BSSS was a comprehensive, nationwide, school-based RE screening initiative conducted as a public health intervention. The BSSS attained a high school coverage rate of 99.5%, screening a total of 164,365 students, representing 98.8% of the school children in Bhutan. A total of 16,376 spectacles were dispensed free of charge as part of this initiative [9].

Despite the successful implementation of the BSSS, comprehensive analysis of the dataset was delayed due to the reallocation of national priorities following the onset of the COVID-19 pandemic, leaving the dataset largely un-examined. A component of the BSSS, using the Refractive Error in School Children (RESC) protocol, was published [10]; however, further integrative analysis combining the RESC-based findings with the broader BSSS data was not undertaken. This secondary analysis of the BSSS data seeks to estimate the national prevalence of RE among children, examine its distribution across demographic and geographic subgroups, and identify

key predictors. Additionally, the analysis aims to compare RESC-derived estimates and census-level findings. The resulting findings are expected to enhance the validation and interpretation of population-based screening methodologies.

Supported by robust, population-level data, the findings of this study are anticipated to inform evidence-based policy formulation for school-based eye health interventions and contribute to Bhutan's advancement towards meeting the WHO eREC target for 2030.

## Methodology

### Study design and setting

This study presents a secondary analysis of the BSSS dataset. The BSSS represents the first nationwide, school-based eye screening program carried out from March to December 2019.

The BSSS was led by the Ministry of Health in collaboration with the Ministry of Education and international partners. It adopted a census-type cross-sectional design to screen all school children for RE and provide corrective spectacles as a public health intervention. This analysis explores previously unreported associations and estimates additional prevalence measures not covered in the original RESC-based report [10]. The BSSS adhered to the standardized RESC Protocol, facilitating comparability with international studies. The principal investigator of the current analysis also served as the lead investigator for the BSSS program.

### Study population and sampling

The BSSS encompassed 586 of 589 schools across all 20 Dzongkhags (districts) and four Thromdes (urban municipalities) in Bhutan, achieving a coverage of 99.5%. A total of 164,365 children aged 5–18 years, representing 94.3% of the school-enrolled population in Bhutan, were examined. The analysis included all children enrolled in schools during the study period who underwent visual acuity screening and refraction as part of the survey. Sampling procedures were not applied, as the BSSS was designed to achieve universal coverage of school-enrolled children.

### Protocol for BSSS

**Survey team.** Six teams were deployed to conduct vision screening, refraction, spectacle dispensing, and data collection. Each team comprised six optometrists and eight ophthalmic technicians. The ophthalmic technicians were responsible for performing vision screenings and recording data, whereas the optometrists conducted clinical assessments and diagnoses.

**Training and quality assurance.** The survey team completed a two-day hands-on training focused on the examination procedures, data collection, and quality assurance protocols. A pilot study was subsequently conducted to familiarize team members with all aspects of the study protocol. Additionally, the PI and study investigators provided continuous supervision and monitoring of fieldwork throughout the survey to ensure strict adherence to the protocol.

**Vision screening and refraction procedure.** The examination protocol followed that described in the previous study [10]. In brief, all participants underwent an initial assessment of uncorrected visual acuity (UCVA) at 6 meters using a Tumbling E chart, administered at temporary stations established within each school. Students with a UCVA of ≤ 6/12 in either eye were subsequently evaluated by an optometrist for both objective and subjective refraction. Cycloplegic refraction was performed using 1% cyclopentolate and measured with a Plusoptix A12R autorefractometer (Plusoptix GmbH, Nuernberg, Germany), followed by subjective refraction to determine the best-corrected visual acuity (BCVA) [10]. Fundoscopy and binocular motor function tests were not conducted. VI was defined as UCVA ≤6/12 in the better eye; RE was defined as VA ≤ 6/12 in either eye improving to ≥6/9 after refraction. Children diagnosed with RE received prescription spectacles, while those whose VA did not improve to ≥6/9 after correction were referred to the nearest ophthalmology center for further assessment and management.

### Definitions

The BSSS utilized the definitions specified in the RESC protocol. UCVA of 6/9 or better in both eyes was classified as normal vision and, while excluded from refraction consideration, was included in the calculation of VI prevalence rates. VI was defined as UCVA of ≤ 6/12 in the better eye. URE was defined as a presenting VA of ≤ 6/12 in either eye that improved to ≥ 6/9 with refraction. Myopia was defined as a spherical equivalent (SE) of ≤ -0.50 diopters (D), hypermetropia as ≥ +2.00 D, and astigmatism as a cylindrical power of ≥ ±0.75 D in either eye.

### Data sources and variables

For this secondary analysis, cleaned datasets were obtained from the Jigme Dorji Wangchuck National Referral Hospital (JDWNRH), Primary Eye Care Program (PECP), Ministry of Health. Anonymized data was obtained on 8th November 2023 after completing all official procedures. Variables included gender, school type (public/private), location (urban/rural), and region (east/central/west). Captured data comprised the total number of students with VI and RE, with RE further classified by gender. Spectacle distribution data was sourced from the PECP. Enrollment records from the Annual Education Statistics of the Ministry of Education were used to validate coverage and ensure representativeness of the sample.

### Statistical analysis

Descriptive statistics were employed to summarize demographic variables and estimate the prevalence of VI and RE, stratified by gender, school type, school location, and geographic region. Bivariate analyses were conducted to examine differences in prevalence across these subgroups. Multivariate logistic regression models were used to identify independent predictors of overall RE and its specific sub-types (myopia, hypermetropia, and astigmatism). Odds ratios (ORs) with corresponding 95% confidence intervals (CIs) were reported, and statistical significance was set at $p < 0.05$. As this was a nationwide census dataset, analyses were conducted at the population level. All statistical analyses were performed using IBM SPSS Statistics (IBM Corp., Armonk, NY, USA), v26.0.

### Ethical considerations

Ethical approval for this study was obtained from the Research Ethics Board of Health (REBH), and administrative clearance from the Policy and Planning Division, Ministry of Health, Bhutan. The REBH also approved this secondary analysis of anonymized data without requiring re-consent.

The BSSS received ethical approval from the REBH with additional authorization granted by the Ministry of Education. The survey was implemented by the Ministry of Health as a public health intervention for school children using standardized eye examination protocols.

In Bhutan, school health activities such as medical or vision screening are considered part of the routine school health program, conducted under the joint authority of the Ministry of Health and the Ministry of Education. Written informed consent was obtained from school authorities, including principals and teachers on behalf of students, as they act *in loco parentis* (in the place of parents) during school hours. Moreover, verbal assent was obtained from all participating students to ensure voluntary participation. Since the procedures involved only non-invasive eye examinations and free spectacle provision, the REBH deemed parental written consent unnecessary, provided that school authorities had granted approval and that ethical oversight was maintained throughout.

## Results

### Survey coverage

The BSSS encompassed 99.5% of the schools in Bhutan, examining 586 out of 589 schools. Among 164,365 children screened, 49% (n = 80,469) were male and 51% (n = 83,896) were female and represented 98.7% of the national school-enrolled population. Three schools with 518 students could not be reached due to bad weather and logistical

challenges. Another 1,427 students were not examined due to absence on the examination day, school withdrawal, or refusal to participate. According to the Population and Housing Census of Bhutan (PHCB 2017), the estimated population of children aged 5–18 years was 174,377. Consequently, the BSSS achieved coverage of 94.3% of the target population in this age group, as shown in Fig 1.

## Prevalence of VI and RE

Among the 164,365 school children screened, 144,414 (87.87%; 95% CI: 87.71-88.03) had normal uncorrected visual acuity (UCVA ≥ 6/9 in both eyes). Overall, 12.14% (95% CI: 11.98-12.30) had VI, with a significantly higher prevalence among females than males (13.16% vs. 11.07%; p<0.01). The prevalence of RE was 11.71% (95% CI: 11.55 -11.86) and accounted for 96.1% of all VI cases. A summary of the visual acuity assessment findings is presented in Table 1.

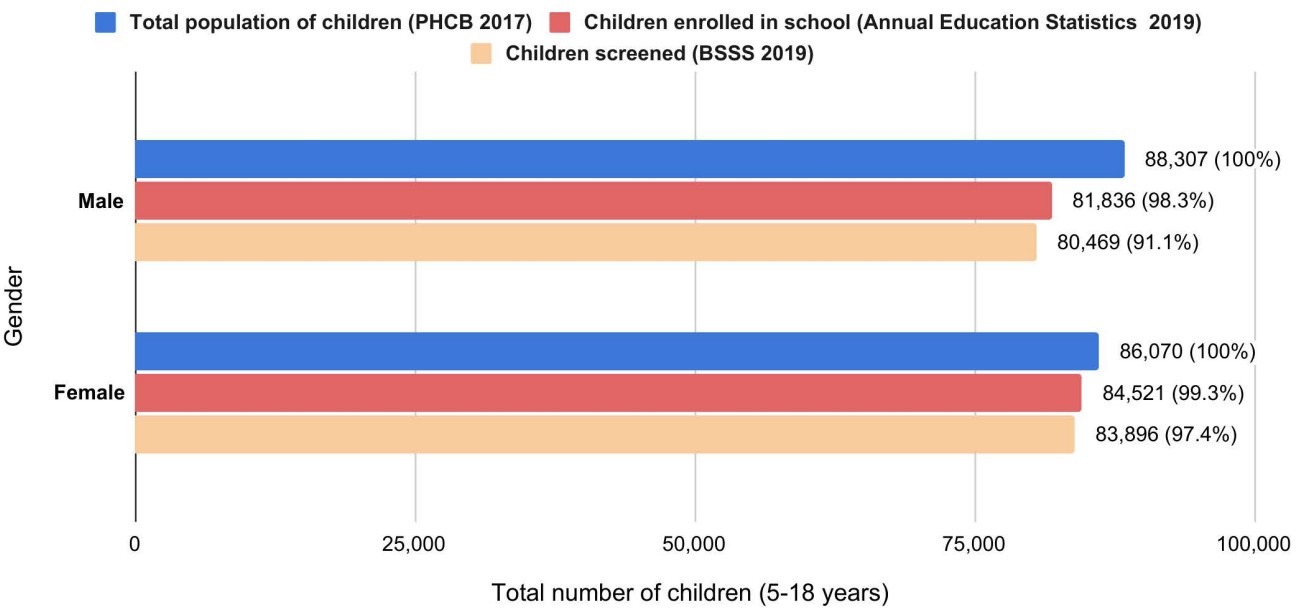

**Fig 1. Coverage of the children during the Bhutan School Sight Survey.**

**Table 1. Prevalence of visual impairment and refractive error among school children in Bhutan.**

| Gender | Total student screened | Total students with normal vision[a] | | Prevalence of VI[b] | | Prevalence of RE[c] | | RE as a cause of VI |
|---|---|---|---|---|---|---|---|---|
| | n | n | % (95%CI) | n | % (95%CI) | n | % (95%CI) | % |
| **Male** | 80,469 | 71,560 | 88.91% (88.69- 89.13) | 8,909 | 11.07% (10.85- 11.29) | 8,563 | 10.64% (10.43- 10.85) | 96.1 |
| **Female** | 83,896 | 72,854 | 86.84% (86.61- 87.07) | 11,042 | 13.16% (12.93- 13.39) | 10,680 | 12.73% (12.50- 12.96) | 96.7 |
| **Total** | 164,365 | 144,414 | 87.87% (87.71- 88.03) | 19,951 | 12.14% (11.98- 12.30) | 19,243 | 11.71% (11.55-11.86). | 96.4 |

[a] UCVA ≥ 6/9 in both eyes.

[b] UCVA ≤ 6/12 in the better eye.

[c] UCVA ≤ 6/12 in either eye improves to ≥ 6/9 with refraction.

## Demographic and geographic distribution of RE

The prevalence of RE was significantly higher among female students (12.73%) than male students (10.64%) (p = 0.014). Similarly, students attending private schools showed a greater prevalence of RE (16.45%) compared with public-school students (11.32%) (p < 0.001). Additionally, a significant urban–rural difference was observed, with urban students (16.19%) having a higher prevalence than rural students (9.56%) (p < 0.001). Regional variation was also significant, with the western region (12.68%) showing the highest prevalence, followed by central (11.78%) and eastern (9.64%) regions (p < 0.001), as presented in Table 2.

The prevalence of RE showed considerable variation across districts. The highest prevalence of RE was observed in urban municipalities: Gelephu (18.58%), Thimphu (17.65%), and Phuentsholing (15.09%). Conversely, the lowest prevalence rates were reported in Samtse (7.84%), Mongar (8.69%), and Samdrup Jongkhar (8.31%). Screening coverage was complete in all districts except for Gasa (76%) and Thimphu Thromde (97.1%). District-specific prevalence of RE among students is presented in S1 Table and Fig 2.

## Pattern of RE types

Myopia was the most common RE (n = 9,497; 5.78%, 95% CI: 5.66-5.92), followed by astigmatism (n = 8,644; 5.26%, 95% CI: 5.15-5.37) and hypermetropia (n = 1,100; 0.67%, 95% CI: 0.63-0.71). The distribution of types of RE by demographic characteristics is presented in Table 3.

## Predictors of RE and its subtypes

Multivariable analysis identified several significant predictors of RE among school children in Bhutan (Table 4). Female students had slightly higher odds of RE than males (AOR = 1.18; 95% CI: 1.07-1.29; p = 0.012). Similarly, students enrolled in private schools (AOR = 1.41; 95% CI: 1.29-1.55) and those in urban areas (AOR = 1.68; 95% CI: 1.58-1.79) were significantly more likely to have RE compared to their counterparts (p < 0.001). Regional differences remained notable, with higher odds in the western and central regions compared to the eastern region. The model showed a good fit (Hosmer–Lemeshow $p = 0.45$) and explained 6.4 % of the variance (Nagelkerke $R^2 = 0.064$).

To further explore type-specific patterns, separate multivariable models were employed for myopia, hypermetropia, and astigmatism (Table 5). Female students had significantly higher odds of myopia (AOR = 1.42; 95% CI: 1.36-1.49) and astigmatism (AOR = 1.22; 95% CI: 1.17-1.29) compared with males. Students attending private schools were more likely to exhibit all three types of RE than those in public schools, with the strongest association noted for hypermetropia (AOR = 1.88; 95% CI: 1.51-2.34). Children from urban areas had slightly higher odds of myopia (AOR = 1.07) and astigmatism (AOR = 1.04) than those from rural areas. Compared with students from the central region, those from the eastern

**Table 2. Prevalence of refractive error among school children in Bhutan by gender, school type, location, and region.**

| Variable | Category | Total Screened (n) | Total with RE (n) | Prevalence % (95% CI) | χ² (df) | p-value |
|---|---|---|---|---|---|---|
| **Gender** | Male | 80,469 | 8,563 | 10.6 (10.4–10.9) | 6.05 (1) | 0.014 |
| | Female | 83,896 | 10,680 | 12.7 (12.5–13.0) | | |
| **School Type** | Public | 151,831 | 17,179 | 11.3 (11.2–11.5) | 175.8 (1) | <0.001 |
| | Private | 12,534 | 2,062 | 16.5 (15.8–17.1) | | |
| **Location** | Rural | 111,101 | 10,617 | 9.6 (9.4–9.7) | 750.3 (1) | <0.001 |
| | Urban | 53,264 | 8,624 | 16.2 (15.9–16.5) | | |
| **Region** | Central | 35,521 | 4,186 | 11.8 (11.5–12.1) | 165.6 (2) | <0.001 |
| | East | 42,162 | 4,063 | 9.6 (9.4–9.9) | | |
| | West | 86,682 | 10,992 | 12.7 (12.5–12.9) | | |

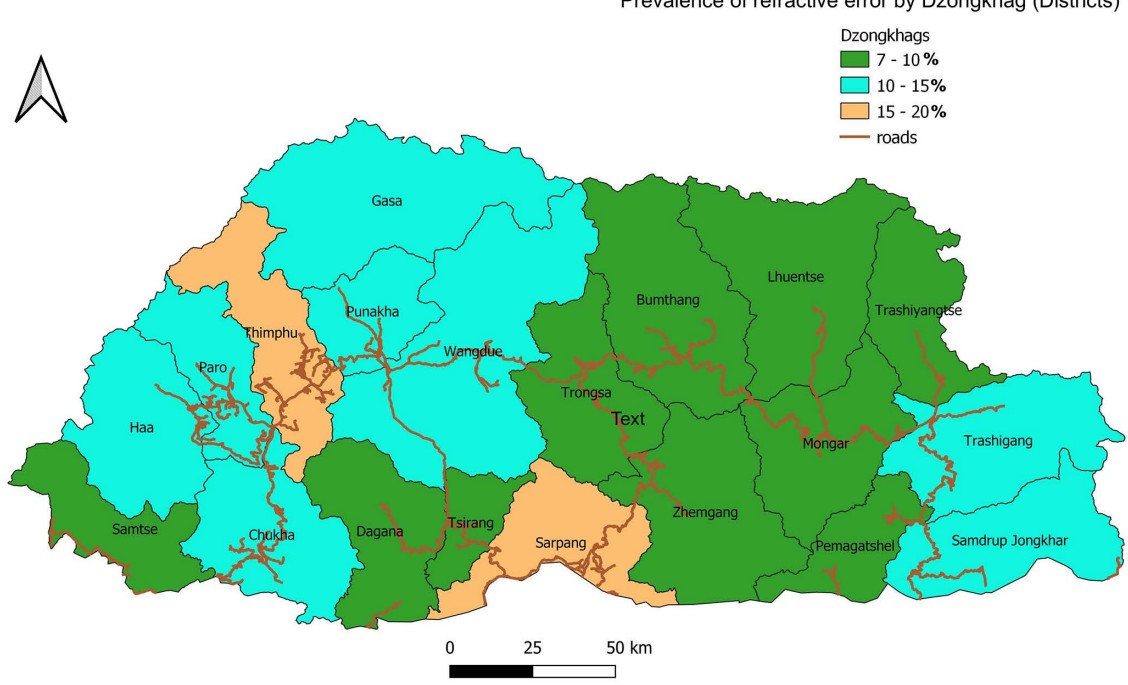

**Fig 2. Geographical distribution of prevalence of refractive error among students by districts in Bhutan.** [PlaniGlobe, http://www.planiglobe.com, CC BY 2.0, spatial processing and map-production were conducted in QGIS 3.16.11 (Hannover) linked with GRASS 7.8.5].

**Table 3. Distribution of types of refractive error by demographic characteristics (n = 164,365).**

| Variable | Category | Total Screened (n) | Myopia n (%) | Hypermetropia n (%) | Astigmatism n (%) | χ² (df) | p-value |
|---|---|---|---|---|---|---|---|
| **Gender** | Male | 80,469 | 3,926 (4.9) | 550 (0.7) | 3,916 (4.9) | 190.4 (2) | <0.001 |
| | Female | 83,896 | 5,571 (6.6) | 550 (0.7) | 4,728 (5.6) | | |
| **School Type** | Public | 151,831 | 8,505 (5.6) | 997 (0.7) | 7,674 (5.1) | 192.6 (2) | <0.001 |
| | Private | 12,534 | 989 (7.9) | 103 (0.8) | 970 (7.7) | | |
| **Location** | Rural | 111,101 | 5,328 (4.8) | 549 (0.5) | 4,740 (4.3) | 510.2 (2) | <0.001 |
| | Urban | 53,264 | 4,169 (7.8) | 551 (1.0) | 3,904 (7.3) | | |
| **Region** | Central | 35,521 | 2,117 (6.0) | 210 (0.6) | 1,959 (5.5) | 112.9 (4) | <0.001 |
| | East | 42,162 | 2,051 (4.9) | 198 (0.5) | 1,814 (4.3) | | |
| | West | 86,682 | 5,329 (6.1) | 692 (0.8) | 4,971 (5.7) | | |

region showed lower odds of both myopia (AOR = 0.86) and astigmatism (AOR = 0.86), while regional differences in hyper-metropia were not significant.

## Spectacle coverage and distribution

Out of the total number of students diagnosed with RE (n = 19,243), only 2213 (11.5%) had appropriate spectacles that improved their visual acuity (VA) to 6/9 or better, resulting in a baseline spectacle coverage of 11.5%. Among the 17,021 students identified as requiring new spectacles during BSSS, 8,595 (50.5%) received ready-to-clip glasses either immediately or within five days following refraction. Of the remaining students, 8,426 (49.5%) needed customized spectacles

**Table 4. Predictors of refractive error among school children in Bhutan using multivariable logistic regression (n = 164,365).**

| Predictor variable | Reference category | Adjusted OR (95% CI) | p-value |
|---|---|---|---|
| Gender | Male | 1.18 (1.07 – 1.29) | 0.012 |
| School type | Public | 1.41 (1.29 – 1.55) | <0.001 |
| Location | Rural | 1.68 (1.58 – 1.79) | <0.001 |
| Region - Central | East | 1.19 (1.10 – 1.29) | <0.001 |
| Region - West | East | 1.28 (1.19 – 1.38) | <0.001 |

**Table 5. Predictors of refractive error subtypes using multivariable logistic regression.**

| Predictor Variable | Category | Myopia AOR (95% CI) | p-value | Hypermetropia AOR (95% CI) | p-value | Astigmatism AOR (95% CI) | p-value |
|---|---|---|---|---|---|---|---|
| Gender | Female | 1.42 (1.36 -1.49) | < 0.001 | 1.04 (0.91-1.18) | 0.55 | 1.22 (1.1-1.29) | < 0.001 |
| Location | Urban | 1.08 (1.03 -1.12) | 0.001 | 1.05 (0.91-1.21) | 0.48 | 1.04 (1.00-1.09) | 0.05 |
| School Type | Private | 1.41 (1.31 -1.52) | < 0.001 | 1.88 (1.51-2.34) | < 0.001 | 1.64 (1.53-1.77) | < 0.001 |
| Region | East | 0.86 (0.80 -0.93) | < 0.001 | 0.88 (0.72-1.07) | 0.2 | 0.86 (0.80-0.92) | < 0.001 |
| | West | 1.04 (0.98 -1.11) | 0.21 | 1.04 (0.88-1.23) | 0.66 | 1.04 (0.98-1.10) | 0.2 |

for astigmatism or high prescriptions. After follow-up, 7,781 (92.3%) spectacles were delivered within four months. Post-program, 16,376 children (96.2%) who needed new spectacles received them . The remaining 645 (3.8%) could not be reached, primarily due to school transfers, dropouts, or completion of grade 12.

## Comparison with RESC study

The BSSS reported a prevalence of VI at 12.14%, whereas the RESC study estimated it at 16.9%. In both studies, RE emerged as the leading cause of VI, accounting for 94.4% and 96.4% of cases, respectively. Female sex and attendance at urban schools were consistently identified as the primary predictors of RE in both investigations. A comparative summary of the methodologies and key findings of the RESC and BSSS is provided in Table 6.

## Discussion

The study provides national-level data on the prevalence of RE among school-aged children in Bhutan, using data from the Bhutan School Sight Survey (BSSS) 2019. The BSSS achieved near-universal coverage, screening 586 of 589

**Table 6. Comparison of methodology and key findings of RESC and BSSS.**

| Components | RESC | BSSS |
|---|---|---|
| Year conducted | March to June 2019 | March to December 2019 |
| Target population | School going children (10–15 years) | All school going children (5–18 years) |
| Sampling method | Cluster sampling of schools across Bhutan | Census method covering all schools |
| Vision screening protocol | RESC Protocol | RESC Protocol |
| Prevalence of Visual Impairment (VA worse than 6/12 in at least one eye) | 16.9% | 12.14% |
| Prevalence of RE | 16% (94.4% of all VI) | 11.71% (96.4% of all VI) |
| Spectacle Coverage | 11.9% | 11.5% |
| Predictors of RE | Female, urban schooling, greater parental education | Female, urban schooling |

schools nationwide (99.5%) and assessing 98.7% of enrolled students, representing 94% of the school-aged population (5–18 years) in Bhutan. This extensive coverage offers substantial evidence to guide national child eye health policy and supports Bhutan's commitment to the WHO SPECS 2030 initiative.

## Prevalence and distribution of RE

The study found that the prevalence of RE among school children in Bhutan was 11.71%. This finding aligns with the estimated pooled prevalence (EPP) reported for South-East Asia, which ranges from 10% to 15%, and is comparable to that reported in Nepal (8.4%) [11,12]. The similarity between Bhutan and Nepal may reflect shared geographic, demographic, and socioeconomic characteristics.

The prevalence of myopia (5.78%), hypermetropia (0.67%), and astigmatism (5.26%) observed in this study aligns with regional estimates reported in Southeast Asia: myopia (4.9%), hypermetropia (2.2%), and astigmatism (9.8%) [11]. Similarly, the EPP of RE in Nepal is 7.1% for myopia, 1.0% for hypermetropia, and 2.2% for astigmatism [12]. The relatively low prevalence of hypermetropia, despite the use of cycloplegic refraction, is consistent with findings from most Asian studies. It may be attributed to the RE definition employed in RESC studies, which excludes mild to moderate hypermetropia (+0.50 to +1.99 D) from prevalence estimates.

Myopia is markedly lower in Bhutan than that reported in East Asian countries such as China (36.6%) and South Korea (51.9%) [13,14]. The differences may be associated with Bhutan's relatively slower pace of urbanization, higher levels of outdoor activity, relatively lower near-work demands, and limited screen time exposure. However, as Bhutan undergoes a socioeconomic transition from a least developed country status, along with increased rural-to-urban migration and greater mobile phone penetration, myopia prevalence is anticipated to increase. Additionally, the emergence of 'left-behind children' who are primarily cared for by grandparents or relatives while parents seek education or employment abroad, may face increased exposure to unsupervised digital device use, potentially exacerbating the rising trend of myopia [15].

The nearly equivalent prevalence of myopia (5.78%) and astigmatism (5.26%) warrants the need for cycloplegic refraction and comprehensive optical services within school screening programs in Bhutan. However, the prevalence of astigmatism observed in this study is relatively lower than that reported in a study in Xinjiang, China, where astigmatism prevalence exceeded 36% [16]. Studies suggest that ready-made spectacles could address the refractive needs of over 80% of children with URE [17]. However, the implementation of ready-to-clip spectacles during the BSSS faced challenges, such as difficulties in maintaining an adequate and diverse stock of lenses and frames at remote screening sites.

## Demographic and geographic disparities

Female students, those in private schools, and urban schooling were significant predictors of RE, consistent with global trends. Female students showed a higher prevalence of RE compared to males. This finding aligns with studies from Asia, which suggest that gender-related disparities in access to eye care and a potentially earlier onset of myopia in females may contribute to the observed differences [18,19]. Students attending private schools exhibited a higher prevalence of RE compared to those in public schools. This difference may reflect lifestyle factors, including increased near-work demands and reduced outdoor exposure, both recognized risk factors for RE [20].

Additionally, notable differences in RE prevalence were observed between students in rural (9.56%) and urban (16.19%) schools. The highest prevalence was reported in urban municipalities, while the lowest prevalence was observed in districts of southern and eastern Bhutan. These variations may be influenced by urbanization and economic growth. However, a recent meta-analysis of myopia prevalence in Indian school children supports these findings while also indicating a significant increase in prevalence among rural children [21].

 

### Spectacle coverage and distribution

While screening and diagnosis coverage was high, spectacle provision remained a major challenge. The low baseline spectacle usage rate (11.5%) highlights persistent access barriers. Although 96.2% of children eventually received spectacles within four months, the delay in delivery and initial lack of correction reflect gaps in service accessibility. This highlights the need to strengthen the systems for timely spectacle delivery and follow-up care, components central to the WHO SPECS initiative [22]. Improving logistics and stock management for both custom and ready-made spectacles will be equally essential. The use of a unique student identification code, provided by the Ministry of Education, facilitated efficient follow-up and spectacle distribution. This emphasizes the importance of robust cross-sectoral collaboration to ensure sustainable and timely service delivery.

### Comparison with BSSS

Compared to the earlier RESC study in Bhutan, which reported a RE prevalence of 16%, this broader census found a lower overall prevalence of 11.7% [10]. This difference is likely due to the broader age range in the BSSS (5–18 years) compared to the RESC (10–15 years). RE prevalence tends to be higher among older children (12–18 years) compared to younger age groups (5–11 years) [19]. The inclusion of lower age groups in BSSS may have resulted in a lower overall prevalence estimate. However, both studies identified female and urban schooling as predictors of RE, confirming the reliability and external validity of the BSSS findings. These findings support the utility of the RESC methodology for national planning and integration of school-based vision screening into broader health and education policies.

### Strengths, limitations, and future research

The study's major strengths are its near-universal coverage, large sample size, and the use of standardized RESC protocols and cycloplegic refraction, which minimizes measurement bias. However, the lack of data on screen time, outdoor activity, parental history, and socioeconomic status limits comprehensive risk assessment. As this was a census-style survey, the statistical analyses describe associations rather than sample-based estimates.

Future studies should adopt longitudinal study designs to monitor the progression of RE and evaluate the outcomes of interventions over time. Additionally, the inclusion of behavioral and socioeconomic variables is essential to enable a more comprehensive understanding of associated risk factors. Studies assessing the impact, compliance, and acceptability of free spectacle provision are also warranted to inform the effectiveness and sustainability of public health interventions.

## Conclusion

This analysis indicates that RE is the leading cause of visual impairment (VI) among school children in Bhutan, affecting slightly over one in ten students. Myopia and astigmatism account for the majority of cases. While the overall prevalence of RE is lower than that reported in neighboring countries, the disproportionately higher burden among females, urban students, and those in private schools highlights persistent inequities in refractive services.

The low baseline spectacle coverage further underscores the disconnect between diagnosis and access to corrective services. These findings align with prior RESC results and demonstrate that large-scale, school-based eye screening programs are both feasible and effective in small countries. BSSS-like interventions could serve as a viable model for implementing national eye health initiatives.

As Bhutan advances toward universal eye health, this study provides essential baseline evidence to inform future policy decisions and guide the development of targeted interventions. Addressing RE in school children will require sustained investment in public health follow-up measures, strengthed integrated school vision programs, timely spectacle provision, and enhanced cross-sectoral collaboration.

## Supporting information

**S1 Table. Prevalence of refractive errors among students in all 20 districts (Dzongkhags) and four municipal towns (Thromde) in Bhutan.**
(DOCX)

**S1 Dataset. Dataset for the Bhutan School Sight Survey.**
(XLSX)

## Acknowledgments

The authors thank Dr. Zelalem G Dessie, University of KwaZulu Natal, South Africa for data analysis. The authors also acknowledge the officials of the Policy and Planning Division, and Jigme Dorji Wangchuck National Referral Hospital, Primary Eye Care Program, Ministry of Health, Bhutan for granting the administrative and technical clearance, and sharing data for use in this study.

## Author contributions

**Conceptualization:** Kovin Shunmugam Naidoo, Khathutshelo Percy Mashige, Nor Tshering Lepcha.

**Data curation:** Indra Prasad Sharma, Kovin Shunmugam Naidoo, Khathutshelo Percy Mashige, Nor Tshering Lepcha.

**Formal analysis:** Indra Prasad Sharma, Khathutshelo Percy Mashige.

**Investigation:** Indra Prasad Sharma, Nor Tshering Lepcha.

**Methodology:** Indra Prasad Sharma, Nor Tshering Lepcha.

**Project administration:** Kovin Shunmugam Naidoo, Khathutshelo Percy Mashige.

**Resources:** Indra Prasad Sharma, Kovin Shunmugam Naidoo, Khathutshelo Percy Mashige, Nor Tshering Lepcha.

**Software:** Indra Prasad Sharma.

**Supervision:** Kovin Shunmugam Naidoo, Khathutshelo Percy Mashige, Nor Tshering Lepcha.

**Validation:** Kovin Shunmugam Naidoo, Khathutshelo Percy Mashige, Nor Tshering Lepcha.

**Visualization:** Indra Prasad Sharma.

**Writing – original draft:** Indra Prasad Sharma, Kovin Shunmugam Naidoo.

**Writing – review & editing:** Indra Prasad Sharma, Kovin Shunmugam Naidoo, Khathutshelo Percy Mashige, Nor Tshering Lepcha.

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
