## [Decision Letter · Decision Letter 0]

12 Oct 2025

PGPH-D-25-02130

Refractive error among school-aged children in Bhutan: A secondary analysis of data from nationwide Bhutan School Sight Survey

Dear Dr. Sharma,

Thank you for submitting your manuscript to PLOS Global Public Health. After careful consideration, we feel that it has merit but does not fully meet PLOS Global Public Health’s publication criteria as it currently stands. Therefore, we invite you to submit a revised version of the manuscript that addresses the points raised during the review process.

Please note that we have only been able to secure a single reviewer to assess your manuscript. We are issuing a decision on your manuscript at this point to prevent further delays in the evaluation of your manuscript. Please be aware that the editor who handles your revised manuscript might find it necessary to invite additional reviewers to assess this work once the revised manuscript is submitted. However, we will aim to proceed on the basis of this single review if possible.

The reviewers comments are available in the attached document. They have requested additional reporting of statistical values to support your claims and further clarification around the consent statement. Please review the comments and make appropriate revisions to address the concerns raised.

We look forward to receiving your revised manuscript.

Kind regards,

Emma Campbell, Ph.D

Staff Editor

Journal Requirements:

Additional Editor Comments (if provided):

Reviewers' comments:

Reviewer's Responses to Questions

**Comments to the Author**

1. Does this manuscript meet PLOS Global Public Health’s publication criteria?

Reviewer #1: Yes

2. Has the statistical analysis been performed appropriately and rigorously?

Reviewer #1: Yes

3. Have the authors made all data underlying the findings in their manuscript fully available (please refer to the Data Availability Statement at the start of the manuscript PDF file)?

Reviewer #1: No

4. Is the manuscript presented in an intelligible fashion and written in standard English?

Reviewer #1: Yes

Reviewer #1: This is an important manuscript describing the magnitude of uncorrected refractive error (URE) in school children albeit using secondary data. The strength of the present manuscript is the national coverage of the data given almost a perfect picture of the burden of URE in Bhutan. Minor corrections suggested in the text.

**Do you want your identity to be public for this peer review?** For information about this choice, including consent withdrawal, please see our Privacy Policy

Reviewer #1: **Yes:**  Dr Godwin Ovenseri-Ogbomo

---

## [Decision Letter · Decision Letter 1]

2 Feb 2026

Refractive error among school-aged children in Bhutan: A secondary analysis of data from nationwide Bhutan School Sight Survey

PGPH-D-25-02130R1

Dear Mr Sharma,

We are pleased to inform you that your manuscript 'Refractive error among school-aged children in Bhutan: A secondary analysis of data from nationwide Bhutan School Sight Survey' has been provisionally accepted for publication in PLOS Global Public Health.

Best regards,

Sheetal Prakash Silal, Ph.D.

Academic Editor

Reviewer Comments (if any, and for reference):

Reviewer's Responses to Questions

**Comments to the Author**

Reviewer #2: All comments have been addressed

publication criteria?

Reviewer #2: Yes

3. Has the statistical analysis been performed appropriately and rigorously?

Reviewer #2: Yes

4. Have the authors made all data underlying the findings in their manuscript fully available (please refer to the Data Availability Statement at the start of the manuscript PDF file)?

Reviewer #2: Yes

5. Is the manuscript presented in an intelligible fashion and written in standard English?

Reviewer #2: Yes

Reviewer #2: (No Response)

**Do you want your identity to be public for this peer review?** For information about this choice, including consent withdrawal, please see our Privacy Policy

Reviewer #2: **Yes:**  Gabriel Kwaku Agbeshie
